

# Variation of $^{13}$C and $^{15}$N enrichments in different plant components of labeled winter wheat (*Triticum aestivum* L.)

Zhaoan Sun[1], Shuxia Wu[2], Biao Zhu[3], Yiwen Zhang[4], Roland Bol[5], Qing Chen[1] and Fanqiao Meng[1]

[1] Beijing Key Laboratory of Farmland Soil Pollution Prevention and Remediation, College of Resources and Environmental Sciences, China Agricultural University, Beijing, China
[2] Institute of Agricultural Resources and Regional Planning, China Academy of Agricultural Sciences, Beijing, China
[3] Institute of Ecology, College of Urban and Environmental Sciences, Key Laboratory for Earth Surface Processes of the Ministry of Education, Peking University, Beijing, China
[4] Dryland-Technology Key Laboratory of Shandong Province, Qingdao Agricultural University, Qingdao, China
[5] Institute of Bio- and Geosciences, Agrosphere Institute (IBG-3), Forschungszentrum Jülich GmbH, Jülich, Germany

Corresponding author
Fanqiao Meng, mengfq@cau.edu.cn

## ABSTRACT

Information on the homogeneity and distribution of $^{13}$carbon ($^{13}$C) and nitrogen ($^{15}$N) labeling in winter wheat (*Triticum aestivum* L.) is limited. We conducted a dual labeling experiment to evaluate the variability of $^{13}$C and $^{15}$N enrichment in aboveground parts of labeled winter wheat plants. Labeling with $^{13}$C and $^{15}$N was performed on non-nitrogen fertilized (−N) and nitrogen fertilized (+N, 250 kg N ha$^{-1}$) plants at the elongation and grain filling stages. Aboveground parts of wheat were destructively sampled at 28 days after labeling. As winter wheat growth progressed, $\delta^{13}$C values of wheat ears increased significantly, whereas those of leaves and stems decreased significantly. At the elongation stage, N addition tended to reduce the aboveground $\delta^{13}$C values through dilution of C uptake. At the two stages, upper (newly developed) leaves were more highly enriched with $^{13}$C compared with that of lower (aged) leaves. Variability between individual wheat plants and among pots at the grain filling stage was smaller than that at the elongation stage, especially for the −N treatment. Compared with those of $^{13}$C labeling, differences in $^{15}$N excess between aboveground components (leaves and stems) under $^{15}$N labeling conditions were much smaller. We conclude that non-N fertilization and labeling at the grain filling stage may produce more uniformly $^{13}$C-labeled wheat materials, whereas the materials were more highly $^{13}$C-enriched at the elongation stage, although the $\delta^{13}$C values were more variable. The $^{15}$N-enriched straw tissues via urea fertilization were more uniformly labeled at the grain filling stage compared with that at the elongation stage.

# INTRODUCTION

Precise quantification of carbon (C) and nitrogen (N) cycling in plant–soil systems requires the use of specialized techniques, particularly for tracking above- and belowground residue C and N dynamics in soil (*Shan et al., 2012*; *Meng et al., 2017*; *Zheng et al., 2018*; *Xu et al., 2019*). This reflects the difficulty of detecting small changes in soil organic C and N against the background of substantial soil C and N pools (*Liang et al., 2013*; *Sun et al., 2018*; *Pausch & Kuzyakov, 2018*). Isotopic labeling of plants using $^{13}$C and $^{15}$N is a powerful tool to track the fate of C and N derived from different plant organs in a soil–plant system when the $^{13}$C- and $^{15}$N-labeled tissues decompose simultaneously during *in situ* (*An et al., 2015*; *Xu et al., 2019*) or laboratory (*Meng et al., 2017*; *Fang et al., 2018*) incubation. However, to draw unbiased conclusions from the fate of isotope-labeled straw C and N, quantitatively tracing the distribution of labeled straw C and N in various soil C and N pools require uniform $^{13}$C- or $^{15}$N-labeled plant materials, because the calculations assume that all plant parts are evenly labeled (*Thompson, 1996*; *Girardin et al., 2009*; *Nguyen Tu et al., 2013*; *Soong et al., 2014*). Therefore, assessment of the variability and the degree of $^{13}$C or $^{15}$N label enrichment among plant organs and tissues is important because the rate of decomposition differs among plant tissues. In addition, the homogeneity of the signal in labeled plant residues may affect quantification of the proportional contribution of soil organic matter and supplementary plant material to plant nutrient uptake in decomposition studies.

Temporal variability in $^{15}$N among N sources in the soil and in allocation of N to different plant organs will result in labeled plant tissues that differ markedly in $^{15}$N contents (*Wagger, Kissel & Smith, 1985*; *Crozier, King & Volk, 1993*). Uniform labeling of tissues is a potential advantage of using a single source of N in a controlled growth system, such as a soil-free culture system supplied with $^{15}$N-fertilizer solution. Homogeneous $^{15}$N enrichment among plant organs can be relatively easy to achieve by fertilizing the plants with a $^{15}$N tracer (*Liang et al., 2013*; *Soong et al., 2014*), whereas attaining uniformly $^{13}$C-labeled crop residues by means of $^{13}$CO$_2$ fixation poses numerous challenges (*Thompson, 1996*; *Girardin et al., 2009*; *Nguyen Tu et al., 2013*; *Soong et al., 2014*). The latter impediment is because rates of C incorporation (photosynthesis) differ widely among plant organs and thus enhances variation in $^{13}$C isotope incorporation in the plant tissues (*Thompson, 1996*; *Girardin et al., 2009*; *Nguyen Tu et al., 2013*; *Soong et al., 2014*). Pulse or continuous labeling of aboveground plant parts has frequently been used to generate $^{13}$C- or $^{14}$C-labeled plant materials (*Zhu & Cheng, 2011*; *An et al., 2015*; *Meng et al., 2017*). Continuous supply of either $^{14}$C or $^{13}$C is advantageous over pulse labeling for uniform labeling of all plant organs, but the procedure requires specialized facilities and is applied from emergence of the first leaf until harvest, therefore it is impossible to apply this approach under field conditions (*Zhu & Cheng, 2011*). Labeling with $^{13}$C (or $^{14}$C) and $^{15}$N under field conditions shows potential for analysis of C and N dynamics derived from crop residues, which are affected by the chemical composition and quantities of the crop residue (*Berg et al., 1991*; *Tahir et al., 2018*). Concentrations of phenolic compounds, for example, may alter decomposition rates (*Fox, Myers & Vallis, 1990*; *Ranells & Wagger, 1992*), may be sensitive to differences in field, greenhouse, or growth-chamber ultraviolet light intensities

(*Crozier, King & Volk, 1993*). Pulse labeling is advantageous over continuous labeling for field isotopic labeling because pulse labeling is easier to execute and requires simple instrumentation (*Berg et al., 1991*; *Tahir et al., 2018*). The majority of labeled crop residues used in previous studies of C and N dynamics were labeled with $^{13}C$ (or $^{14}C$) and $^{15}N$ tracers under controlled conditions.

To produce labeled crop residues, previous field"- labeling experiments mainly focused on the degree of $^{13}C$ and $^{15}N$ enrichment in crop residues (*Berg et al., 1991*; *Crozier, King & Volk, 1993*; *Tahir et al., 2018*). However, few researchers evaluated the homogeneity and distribution of $^{13}C$ and $^{15}N$ labels within the labeled plant tissues, especially in residues of winter wheat, which is among the most important staple crops worldwide and is frequently used in labeling studies (*Jia et al., 2011*; *Butterly et al., 2015*; *Chen et al., 2016*; *Sun et al., 2018*). In annual cereal crops, the plants grow vigorously and enrichment of labeled C is higher during the juvenile stage than at maturity (*Meng et al., 2013*; *Liu, Jiang & Li, 2015*; *Sun et al., 2018*). The majority of C labeling studies have been conducted in the early developmental stages of wheat, e.g., continuous labeling 60 days after emergence (*Liljeroth, Kuikman & Van Veen, 1994*; *Marx et al., 2007*) and pulse labeling <150 days after emergence (*Martens et al., 2009*; *Butterly et al., 2015*). Few studies have compared the enrichment of labeled C for multiple labeling periods. In addition, heavy application of N fertilizer delays senescence in wheat, which results in a substantial quantity of non-structural carbohydrates remaining in the straw and affects photosynthesis (*Yang & Zhang, 2006*), may lead to changes in the degree of labeled C enrichment in straw.

To evaluate the uniformity of labeling within plants, in the present study the shoot of an individual plant of winter wheat was divided into leaves and the stem from three sections of equal length using the method described by *Thompson (1996)*. We conducted a $^{13}C$ pulse-labeling and $^{15}N$ fertilization experiment for labeling pot-grown winter wheat under field conditions at different growth stages. The objectives were (i) to assess the homogeneity and degree of $^{13}C$ and $^{15}N$ enrichment among different organs and among identical organs (stems and leaves) in different portions of the shoot (upper, middle, and lower); and (ii) to investigate the impacts of wheat growth and N fertilization on isotope variability among aboveground compartments after $^{13}C$ labeling.

## MATERIALS AND METHODS

### Treatments and $^{15}N$ labeling procedure

In the current study, we used winter wheat (*Triticum aestivum* L. 'Luyuan 502'), a staple crop grown worldwide (*Zhao et al., 2016*). To examine the impacts of wheat growth stage and N fertilization on variation in $^{13}C$ enrichment of winter wheat, two N treatments (0 mg N kg$^{-1}$ soil, −N; and 90 mg N kg$^{-1}$ soil, equivalent to 250 kg N ha$^{-1}$, +N) were applied, and wheat plants grown in a calcareous soil (fluvo-aquic sandy loam) were pulse-labeled with $^{13}CO_2$ at the elongation (168 days after sowing; DAS) and grain filling (205 DAS) stages (Figs. 1A and 1B). To examine the degree of variation in $^{15}N$ enrichment among aboveground compartments, $^{15}N$-labeled fertilizer ($^{15}N$-labeled urea; Shanghai Research Institute of Chemical Industry, Shanghai, China) was applied by direct addition to the soil

 

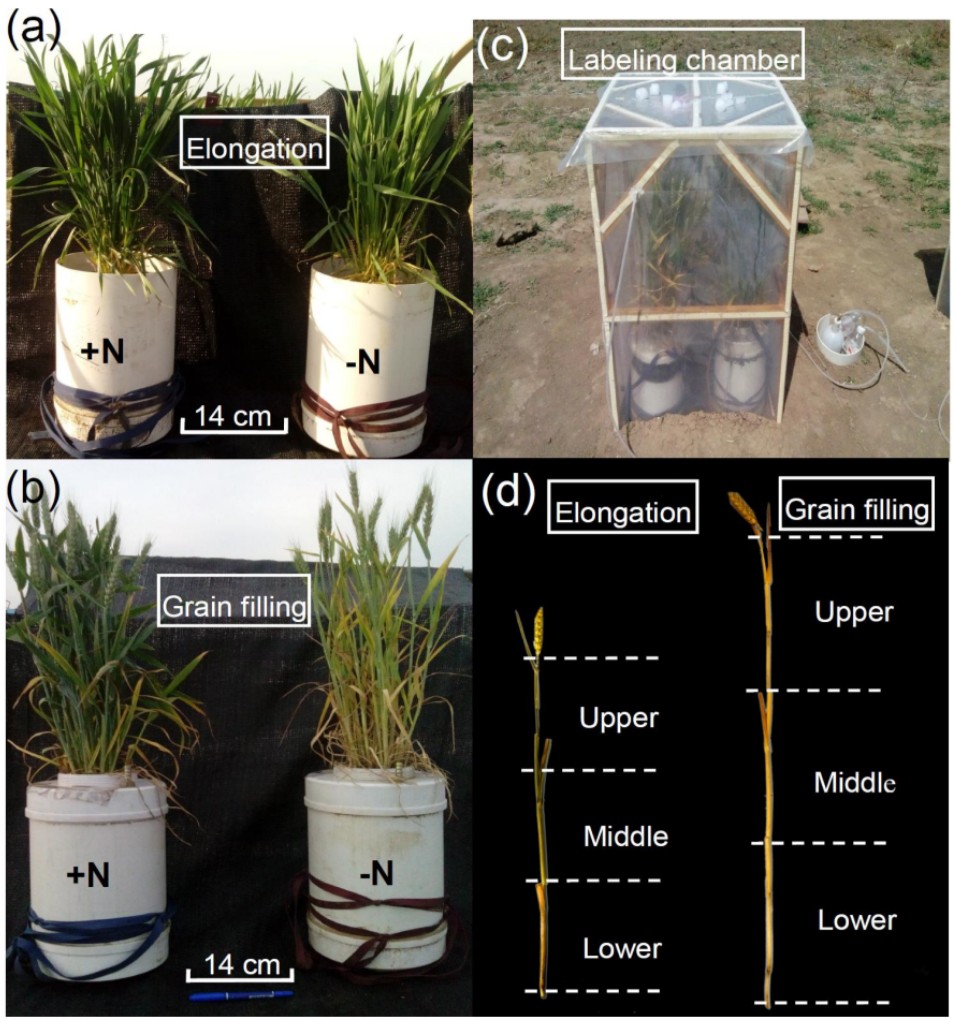

**Figure 1** Winter wheat growth 168 days after sowing (elongation stage) (A) and 205 days after sowing (grain filling stage) (B). Field labeling chamber for winter wheat (C). Schematic diagram of the three sampled portions of the shoots (D). −N = no nitrogen addition; +N = nitrogen addition.

for the +N treatment ($^{15}N$ 10.14 atom %; about US$10 per gram of $^{15}N$ labeled urea). Additional wheat plants were selected as $^{15}N$ natural abundance controls and were fertilized with unlabeled urea as for +N treatment. Ten pre-germinated (24–48 h) winter wheat seeds were sown directly into PVC pots (30 cm ×20 cm i.d.) containing 8.7 kg air-dried soil, and then the soil was moistened to 70% water-holding capacity. One week after germination (20 October, 2014), all except the six strongest seedlings were removed from each pot (equivalent to a field planting density of 1.5 million plants ha$^{-1}$). The experiment was conducted under field conditions. For the +N treatment, half of the $^{15}N$ labeled or unlabeled urea was applied as a basal dressing at sowing (0.84 g $^{15}N$ labeled or unlabeled urea was applied to each pot) and the remaining half was used to topdress plants at the elongation stage (15 March, 2015). The wheat plants for the first $^{13}CO_2$ labeling were fertilized as a single basal application and received 45 mg N kg$^{-1}$ air-dried soil. The wheat

plants for the second $^{13}CO_2$ labeling received a total of 90 mg N kg$^{-1}$ air-dried soil in two applications during the experiment (1.68 g $^{15}$N labeled or unlabeled urea was applied to each pot, which corresponded to a total fertilization of 250 kg N ha$^{-1}$), which was split into $2 \times 45$ mg N kg$^{-1}$air-dried soil applications (basal urea N and topdressing urea N). Monopotassium phosphate (KH$_2$PO$_4$) fertilizer was applied as a basal dressing at the rate 17 mg P and 18 mg K kg$^{-1}$ air-dried soil, which conformed with the local agricultural practice. To simulate plant litter similar to that occurring under the field-grown conditions in this study, we applied $^{15}$N-urea to the soil of pot-grown winter wheat in the field rather than supplying $^{15}$N-fertilizer solution to a soil-free culture system.

## Growth conditions and $^{13}$C pulse-labeling procedure

Soil was collected from the plough layer (0–20 cm) of farmland near the Agro-Ecosystem Experimental Station of China Agricultural University, Huantai County, Shandong Province (36°57′N, 117°59′E, 18 m elevation). The soil of the experimental field was derived from Yellow River alluvial sediments and was classified as an Aquic Inceptisol (a calcareous, fluvo-aquic sandy loam; *Shi et al., 2013*). Soil properties were as follows: soil organic $C = 8.4$ g kg$^{-1}$, soil inorganic $C = 5.2$ g kg$^{-1}$, total $N = 0.67$ g kg$^{-1}$, NH$_4$-N $= 1.6$ mg kg$^{-1}$, NO$_3$ $-N = 17.2$ mg kg$^{-1}$, pH $= 8.2$ (soil:water $= 1:2.5$), available $K = 174$ mg kg$^{-1}$, and Olsen $P = 5.2$ mg kg$^{-1}$. After sampling, the soil was air-dried, homogenized, and sieved using a five mm screen prior to the experiment. Each pot (height $= 30$ cm; inner diameter $= 20$ cm) was filled with 8.7 kg of air-dried soil to a bulk density of 1.38 g cm$^{-3}$. Wheat roots were mainly concentrated in the 0–20 cm soil layer and contributed 57.6%–78.6% of the 0–100 cm total root weight in typical farmland on the North China Plain (*Hou, 2011*). Hence, pots with 30 cm soil depth were considered sufficient for normal wheat root development in this study. The pot-grown winter wheat seedlings were transferred to an experimental field at the Agro-Ecosystem Experimental Station. The pots containing wheat plants were placed in cropland holes (30 cm depth; 24 cm diameter) to simulate local wheat-growing conditions. The total growth period was 230 days, with six developmental stages recognized: (i) seeding (0–17 DAS), (ii) tillering (18–150 DAS), (iii) elongation (151–179 DAS), (iv) anthesis (180–193 DAS), (v) grain filling (194–214 DAS), and (vi) dough ripening (215–230 DAS). Precipitation was 147 mm and the soil temperature ranged from $-1.6$ to 29.7 °C during the wheat growing season (*Zhao et al., 2017*). The soil water content of each pot, which was controlled gravimetrically to simulate local wheat production, was adjusted daily to 65% (during the seedling stage), 70% (tillering), 80% (elongation), 80% (anthesis), and 70%–75% (grain-filling) of the field water-holding capacity in accordance with the amount of rainfall and evaporation.

Labeling with $^{13}CO_2$ was performed at the elongation and grain filling stages (i.e., 168 and 205 DAS). On each occasion, four pots were randomly selected for $^{13}CO_2$ labeling from each of the −N and +N treatments, and an additional eight pots were selected as $^{13}CO_2$ unlabeled controls and were maintained separately from the labeled plants. Hence, we used 32 pots in the experiment, with three replicates for each N treatment, for destructive sampling at 28 days after labeling during each of the developmental stages (three from four pots per sampling for each treatment): eight pots were dual labeled with $^{13}$C and

$^{15}$N in the +N treatment; eight pots were labeled with $^{13}$C for the −N treatment; and an additional 16 pots were selected as unlabeled controls (eight pots for −N and eight pots for +N). We considered each pot to be a technical replicate for each treatment (three from four pots per sampling for each treatment), and took each plant within a pot to be a biological replicate (three plants chosen within the pot). A chamber (0.6 m long ×1.0 m high; Fig. 1C), adapted from *Swinnen, Van Veen & Merckx (1994)* and *Meng et al. (2013)*, was used for $^{13}CO_2$ labeling. On the previous day, the soil surface was covered with a PVC board and sealed with silicon, including around the stems. To remove all $^{12}CO_2$ before labeling, the air inside the chamber was continuously circulated for 30 min through 10 M NaOH solution using a membrane pump. The $^{13}$C-labeled $Na_2^{13}CO_3$ at 98 atom % $^{13}$C (about US$100 per gram of $Na_2^{13}CO_3$) was provided by the Shanghai Research Institute of Chemical Industry. A beaker containing $Na_2^{13}CO_3$ (8.0 g $Na_2^{13}CO_3$ for each labeling event) was placed inside the chamber. To determine the $CO_2$ concentration within the chamber using an infrared gas analyzer (CI301PS; CID Bio-Science, Camas, WA, USA), an unlabeled control treatment with an additional four pots was set up in a separate chamber under identical conditions except that $Na_2CO_3$ was used to produce non-labeled $CO_2$. This arrangement was used because of the different wavelengths for maximum absorption of $^{13}CO_2$ and $^{12}CO_2$ in the infrared detector (*Meng et al., 2013*). If the rate of decline in $CO_2$ concentration slowed considerably (below 200 $\mu$L L$^{-1}$) in the chamber containing the unlabeled control treatment, 1 M $H_2SO_4$ was injected until the $CO_2$ concentration increased to approximately 360 $\mu$L L$^{-1}$. The same volume of $H_2SO_4$ solution was also injected into the chamber housing the plants undergoing labeling. A fan was used to homogenize the atmosphere within each chamber. After 7 h of labeling, plants were removed from the chamber to prevent re-assimilation of shoot-respired $^{13}CO_2$.

## Sampling and chemical analyses

The $^{13}$C-labeled winter wheat plants and soils were destructively sampled 28 days after each labeling. No significant difference between the mean aboveground biomass and root biomass of the $^{13}$C pulse-labeled plants and the unlabeled control plants was observed when they were harvested 28 days after labeling during each of the developmental stages. This observation indicated that $^{13}$C pulse labeling did not affect plant growth. To examine the uniformity of labeling within a wheat plant, individual wheat plants were collected at the final harvest and the shoots (stem and leaves) were divided into three equal-length portions (upper, middle, and lower portions; Fig. 1D). Separate analyses of $^{13}$C and $^{15}$N enrichment were undertaken on the leaf and stem portions as well as the ears. Three plants per pot were assessed for variation in $^{13}$C and $^{15}$N enrichment. Shoots (stems and leave) and ears were oven-dried at 60 °C to a constant weight. All samples were ground to a fine powder (<500 $\mu$m) in a ball mill (Restol MM2000, Retsch, Haan, Germany) prior to analysis.

The $^{13}$C and $^{15}$N values of wheat plant materials were determined using isotope ratio mass spectrometry (DELTAplus XP, ThermoFinnigan, Bremen, Germany). The abundance of $^{13}$C was expressed as parts per thousands (‰) relative to the international standard (Pee Dee Belemnite; 0‰) expressed as delta units ($\delta$) (*Craig, 1953*). The $^{15}$N enrichment
of $^{15}$N-labeled plant parts fertilized with labeled urea was reported as atom % $^{15}$N excess calculated by subtracting the values for the $^{15}$N natural abundance control samples fertilized with unlabeled urea (*Coplen, 2011*).

## Data processing and statistical analysis

To estimate the weight-average $^{13}$C and $^{15}$N values of the aboveground parts and three equal-length portions (upper, middle, and lower portion) of the shoot of winter wheat plants, a simple isotopic mass balance mixing equation was used (*Thompson, 1996*):

$$\delta^{13}C_{abveground} = \delta^{13}C_{ear} \times F_{ear} + \delta^{13}C_{leaves} \times F_{leaves} + \delta^{13}C_{stem} \times F_{stem} \tag{1}$$

$$^{15}N_{abveground} = {}^{15}N_{ear} \times F_{ear} + {}^{15}N_{leaves} \times F_{leaves} + {}^{15}N_{stem} \times F_{stem} \tag{2}$$

$$1 = F_{ear} + F_{leaves} + F_{stem} \tag{3}$$

where $\delta^{13}C_{aboveground}$, $\delta^{13}C_{ear}$, $\delta^{13}C_{leaves}$, and $\delta^{13}C_{stem}$ is the C isotope value ($\delta$) for aboveground parts, ear, leaves, and stem of winter wheat, respectively; $^{15}N_{aboveground}$, $^{15}N_{ear}$, $^{15}N_{leaves}$, and $^{15}N_{stem}$ is the $^{15}$N excess (in atom %) for aboveground parts, ear, leaves, and stem of winter wheat, respectively; and $F_{ear}$, $F_{leaves}$, and $F_{stem}$ is the total aboveground dry weight of winter wheat plants in the ear, leaves and stem fractions, respectively.

$$\delta^{13}C_{i\,portion} = \delta^{13}C_{i\,leaves} \times F_{i\,leaves} + \delta^{13}C_{i\,stem} \times F_{i\,stem} \tag{4}$$

$$^{15}N_{i\,portion} = {}^{15}N_{i\,leaves} \times F_{i\,leaves} + {}^{15}N_{i\,stem} \times F_{i\,stem} \tag{5}$$

$$1 = F_{i\,leaves} + F_{i\,stem} \tag{6}$$

where $\delta^{13}C_{i\,portion}$ and $^{15}N_{i\,portion}$ are the C isotope value ($\delta$) and $^{15}$N excess (in atom %) for the upper, middle, and lower portion, respectively, of the shoot of winter wheat plants; and $F_{i\,leaves}$ and $F_{i\,stem}$ are the respective leaves and stem fractions in the same portion of the shoot.

Statistical analysis was carried out with SPSS (version 11.0, 2002; SPSS, Chicago, IL, USA). Two-way analysis of variance was performed to analyze the effect of N fertilization and/or developmental stage on $^{13}$C and $^{15}$N distribution in different aboveground components of the winter wheat plants. The least significant difference (LSD) test was used to examine differences in $^{13}$C and $^{15}$N isotope values between the N treatments or different aboveground components. The significance level of 0.05 was chosen to indicate statistical significance.

## RESULTS

### Winter wheat biomass and total C and N contents

The interactive effect of plant age and N fertilizer treatment on increasing the biomass of the ear and whole plant was significant (Table 1 and Fig. 2A), which indicated that the effect of N fertilization tended to be stronger as wheat growth progressed. Straw, ear, and whole-plant biomass were significantly higher in the +N treatment than in the −N at the elongation and grain filling stages (Fig. 2A). With progression of growth, the C and N contents of wheat straw and the ear significantly increased (Table 1 and Figs. 2B, 2C),

**Table 1  P-values for two-way analysis of variance of the effects of stage (S) and nitrogen (N) on biomass, carbon content, and N content of different winter wheat components.**

| Treatment | Biomass | | | | C content | | | N content | | |
|---|---|---|---|---|---|---|---|---|---|---|
| | Root | Straw | Ear | Total plant | Root | Straw | Ear | Root | Straw | Ear |
| S | 0.886 | <0.001 | <0.001 | <0.001 | 0.598 | 0.003 | <0.001 | 0.282 | <0.001 | <0.001 |
| N | 0.136 | 0.027 | <0.001 | <0.001 | 0.118 | <0.001 | 0.947 | 0.282 | <0.001 | <0.001 |
| S × N | 0.393 | 0.331 | <0.001 | 0.016 | 0.528 | 0.456 | 0.046 | 0.310 | 0.935 | <0.001 |

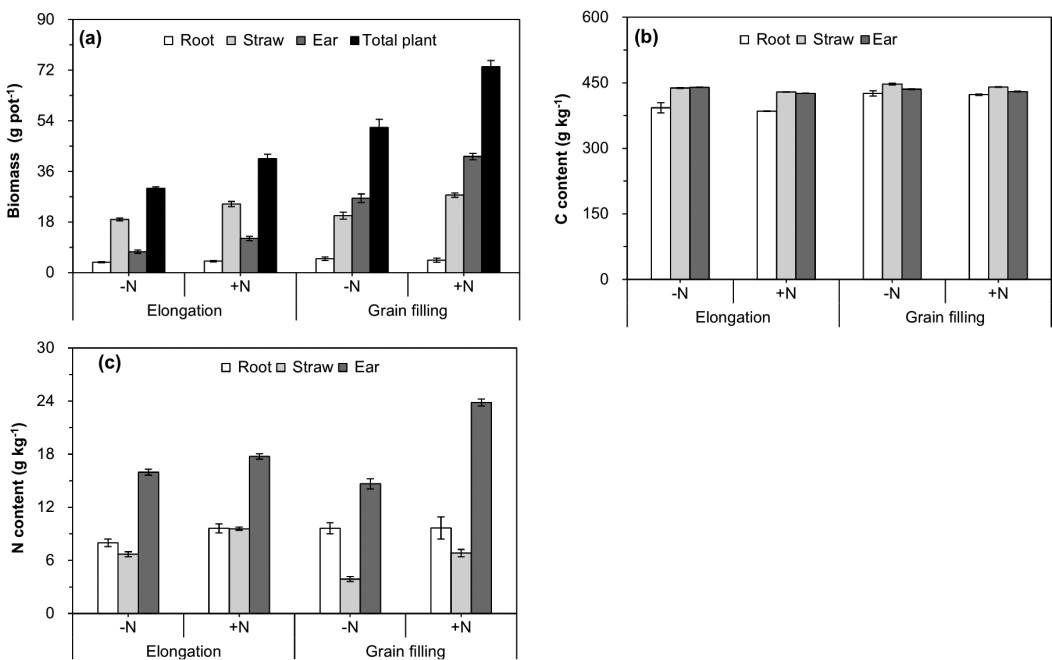

**Figure 2  Biomass (A), carbon content (B), and nitrogen content (C) of different winter wheat components.** Error bars indicate standard errors ($n = 4$). −N = no N addition; +N = N addition.

whereas no significant differences in the roots were observed from the elongation to the grain filling stages (Table 1 and Fig. 2A). Compared with those of the −N treatment, N fertilization significantly decreased the C content of straw, and significantly increased the N content of wheat straw and the ear at the elongation and grain filling stages (Table 1 and Figs. 2B, 2C).

### $\delta^{13}$C values of aboveground components

In the $^{13}$C-labeled wheat plants, the highest $\delta^{13}$C values were observed in leaves, followed by stems and ears at the elongation stage. However, significantly higher $^{13}$C enrichment was observed in the ears compared with the other aboveground parts at the grain filling stage (Table 2). As growth progressed, the $\delta^{13}$C values of ears significantly increased by 130% and 170% for the −N and +N treatments, respectively. In contrast, $\delta^{13}$C values of leaves and stems significantly decreased by 82.3% (−N) and 72.2% ( +N) in leaves and by 72.4% (−N) and 65.5% (+N) in stems. These results indicated that growth stage had a

**Table 2** $\delta^{13}$C value and $^{15}$N excess (in atom %) of different winter wheat components. Values are the mean ± standard error ($n = 9$). Different lowercase letters in the same row denote a significant difference (LSD, $p < 0.05$) in $\delta^{13}$C or $^{15}$N excess (in atom %) between components.

| | | | Ear | Leaves | Stem | Aboveground |
|---|---|---|---|---|---|---|
| $\delta^{13}$C (‰) | Elongation | −N[†] | 106.7 ± 14.8c | 594.9 ± 29.9a[*] | 226.9 ± 15.4b | 295.0 ± 15.3[*] |
| | | +N[‡] | 107.2 ± 18.5c | 375.4 ± 25.0a | 198.8 ± 20.1b | 212.0 ± 16.8 |
| | Grain filling | −N | 249.1 ± 12.1a | 105.7 ± 4.6b | 62.4 ± 1.0c | 81.2 ± 3.1 |
| | | +N | 285.3 ± 11.4a | 103.7 ± 6.2b | 68.5 ± 4.1c | 85.9 ± 5.8 |
| $^{15}$N excess (in atom %) | Elongation | | 3.8 ± 0.2a | 3.71 ± 0.2a | 3.6 ± 0.2a | 3.6 ± 0.1 |
| | Grain filling | | 5.4 ± 0.3a[*] | 4.8 ± 0.1b[*] | 4.9 ± 0.1b[*] | 4.9 ± 0.1[*] |

**Notes.**

[*]in the same column denote either significant differences ($t$-test, $p < 0.05$) in $\delta^{13}$C values between nitrogen (N) treatments at the same stage or significant differences ($t$-test, $p < 0.05$) in $^{15}$N excess (in atom %) between different stages under the same N treatment ($^{15}$N labeling).

[†]−N = no N addition.

[‡]+N = N addition.

**Table 3** $P$-values for two-way analysis of variance of the effects of stage (S) and nitrogen (N) on $\delta^{13}$C value and $\delta^{15}$N excess (in atom %) in different winter wheat components.

| | | Ear | Leaves | | | | Stem | | | |
|---|---|---|---|---|---|---|---|---|---|---|
| | | | Upper | Middle | Lower | Whole | Upper | Middle | Lower | Whole |
| $\delta^{13}$C | N | 0.214 | 0.929 | <0.001 | 0.025 | <0.001 | 0.116 | 0.595 | 0.076 | 0.398 |
| | S | <0.001 | <0.001 | <0.001 | 0.006 | <0.001 | 0.058 | <0.001 | <0.001 | <0.001 |
| | N ×S | 0.225 | 0.635 | 0.000 | 0.077 | <0.001 | 0.545 | 0.952 | 0.199 | 0.192 |
| $^{15}$N excess(in atom %) | S | 0.000 | <0.001 | <0.001 | 0.278 | <0.001 | <0.001 | <0.001 | 0.001 | <0.001 |

significant impact on $^{13}$C distribution among ears, leaves, and stems (Table 3). Nitrogen fertilization decreased $\delta^{13}$C values of aboveground components at the elongation stage and had no significant effect at the grain filling stage; however, the change was only significant in leaves at the elongation stage (36.9%; Tables 2 and 3).

Shoots (leaves and stems) of single plants were divided into three equal-length portions (upper, middle, and lower), to analyze the $^{13}$C distribution within the shoot. Following pulse labeling at the elongation stage, $\delta^{13}$C values of leaves and stems in the different portions showed significant differences. The leaves of the lower portion was the least $^{13}$C enriched for the +N and −N treatments, and $^{13}$C enrichment was highest in the middle and upper portions under the −N and +N treatments, respectively. However, $\delta^{13}$C values of stems were highest in the lower portion, followed by the middle and upper portions, for the −N and +N treatments (Fig. 3). A significant difference was observed at the grain filling stage: for both leaves and stems, $\delta^{13}$C values were significantly higher in the upper portion, with no significant differences between the middle and lower portions observed (Fig. 3).

When we combined $\delta^{13}$C values from the same portions of leaves and stems at the elongation stage, no significant differences were observed among the different portions for the +N treatment, and $\delta^{13}$C values of the upper and middle portions were significantly higher than that of the lower portion for the −N treatment. At the grain filling stage, the $\delta^{13}$C values of the upper portion was highest, followed by those of the middle and lower portions under the −N and +N treatments (Table 4).

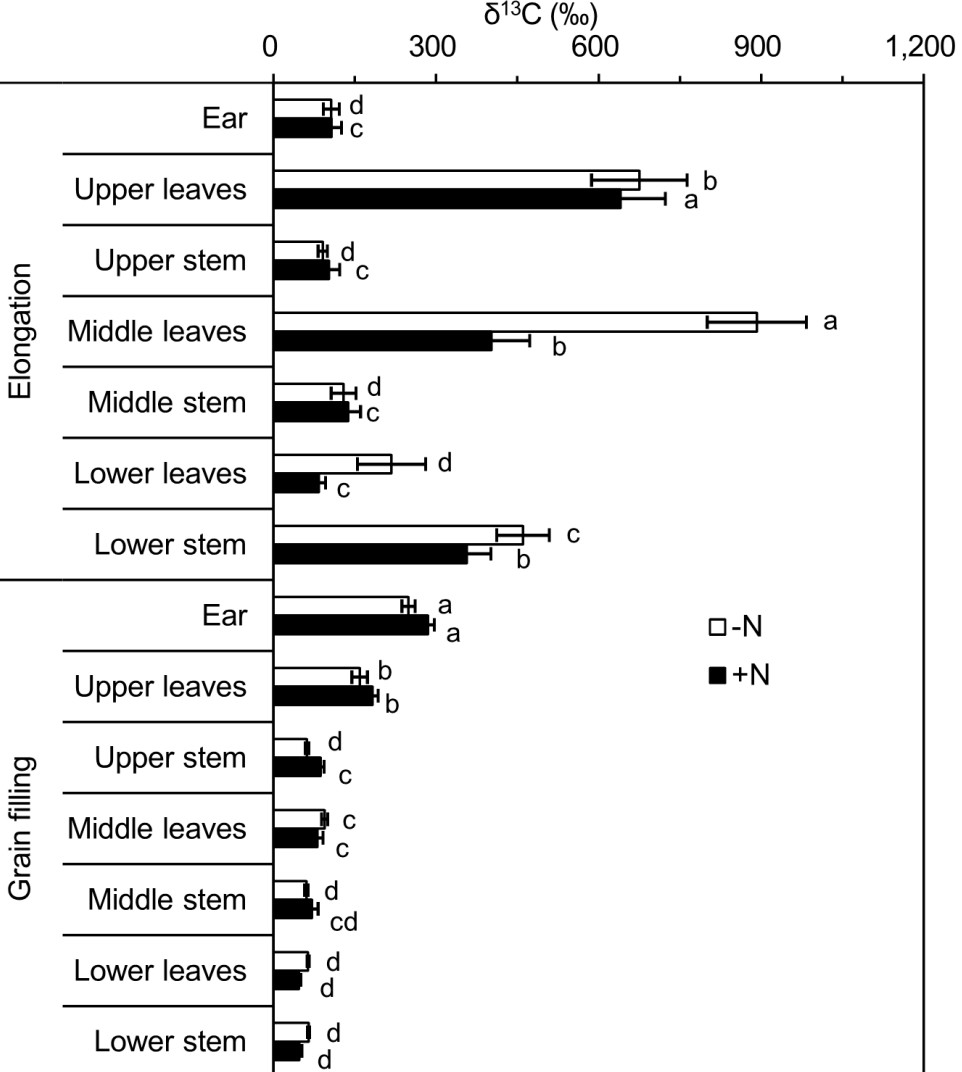

**Figure 3** $\delta^{13}C$ (‰) **values of different winter wheat components.** Different lowercase letters denote a significant difference (LSD, $p < 0.05$) among components at the same developmental stage. Error bars indicate the standard error ($n = 9$). −N = no nitrogen addition; +N = nitrogen addition.

Labeled plants showed coefficients of variation (CVs) for intra-pot (between individual plants) $\delta^{13}C$ of 9.4%–76.4% and 6.3%–31.5% at the elongation and grain filling stages, respectively. Inter-pot CVs (among plants across different pots) for $\delta^{13}C$ ranged from 16.5%–59.0% at the elongation stage and 3.8%–70.9% at the grain filling stage (Table 5). Generally, CVs at the grain filling stage were lower than those at the elongation stage, and N addition tended to increase $\delta^{13}C$ variation, particularly at the grain filling stage.

## $^{15}N$ enrichment variation in aboveground components of winter wheat

Following $^{15}N$-labeling, the rank order of $^{15}N$ excess (in atom %) of ears, leaves, and stems was similar at the elongation and grain filling stages, with $^{15}N$ excess highest in ears,

**Table 4** **Isotope values of upper, middle, and lower portions of shoots in winter wheat.** Values are the mean ± standard error ($n = 9$). Different lowercase letters in the same row denote significant differences in $\delta^{13}C$ or $^{15}N$ excess (in atom %; LSD, $p < 0.05$) between the three portions.

| | | | Upper portion | Middle portion | Lower portion |
|---|---|---|---|---|---|
| $\delta^{13}C$ (‰) | Elongation | −N[†] | 383.0 ± 44.6ab | 510.6 ± 48.2a[*] | 339.1 ± 37.9b[*] |
| | | +N[‡] | 371.2 ± 48.6a | 270.0 ± 38.4ab | 220.1 ± 18.9a |
| | Grain filling | −N | 110.4 ± 7.5a | 77.4 ± 3.5b | 64.3 ± 1.5b |
| | | +N | 134.9 ± 6.9a | 76.1 ± 9.8b | 47.3 ± 3.2c |
| $^{15}N$ excess (in atom %) | Elongation | | 3.76 ± 0.20a | 3.50 ± 0.18a | 3.70 ± 0.20a |
| | Grain filling | | 5.29 ± 0.15a[*] | 4.98 ± 0.19a[*] | 4.42 ± 0.11a[*] |

**Notes.**
[*]in the same column denote either significant differences (t-test, $p < 0.05$) in $\delta^{13}C$ between nitrogen (N) treatments at the same stage or significant differences (t-test, $p < 0.05$) in $^{15}N$ excess (in atom %) between different stages under the same N treatment ($^{15}N$ labeling).
[†]−N = no N addition.
[‡]+N = N addition.

followed by stems and leaves (Table 2). With regard to the three shoot portions (without ears), combined $^{15}N$ excess of leaves and stems were similar among different portions at the elongation and grain filling stages (Table 4). The $^{15}N$ excess of stems and leaves was similar at the elongation stage, but was highest in the upper portion and lowest in the lower portion at the grain filling stage (Fig. 4). As growth progressed, i.e., with increase in biomass, $^{15}N$ excess increased by 32.1%–41.8% (Table 2). Developmental stage had a significant impact on $^{15}N$ excess for all shoot components except leaves in the lower portion (Table 3). Labeled wheat exhibited an intra-pot CV of 7.7%–18.1% at the elongation and grain filling stages; the inter-pot CV was 1.7%–23.4% at the elongation stage, with considerably less variation at the grain filling stage (0.1%–6.2%; Table 5). At both stages, the variation in $^{15}N$ excess was less than that of $\delta^{13}C$. For both $^{15}N$ excess and $\delta^{13}C$ values, intra-pot and inter-pot CVs were comparable, which implied that the degree of variation in the $^{13}C$ and $^{15}N$ distributions was similar.

## DISCUSSION

### Variation in $^{13}C$ enrichment

Average total $\delta^{13}C$ values of aboveground parts of winter wheat plants, resulting from $^{13}CO_2$ release from $Na_2^{13}CO_3$ (98 atom % $^{13}C$) in the field labeling chamber, ranged from 81‰ to 295‰ (Table 2). Given that the majority of previous studies that involved pulse labeling of wheat used $^{14}C$ rather than $^{13}C$, we compared our findings with studies that used $^{13}C$ labeling in other plant species. By injection of $^{13}CO_2$ gas (90 atom% $^{13}C$) in a field-labeling chamber, *Tahir et al. (2018)* achieved a high degree of $\delta^{13}C$ enrichment in wheat with overall $\delta^{13}C$ enrichment of 493‰ in leaves and 357‰ in stems. *Meng et al. (2013)* enriched maize shoots to approximately 125‰–178‰ via $^{13}CO_2$ release from $Ba^{13}CO_3$ (98 atom % $^{13}C$) in a laboratory labeling chamber. The direct $^{13}CO_2$ labeling method has also been used to generate highly $^{13}C$-enriched crop plants, such as rice (533‰–927‰) (*Liu, Jiang & Li, 2015*) and clover (>1,000‰) (*Thompson, 1996*). The $^{13}C$ enrichment of plant materials is dependent on the net $^{12}C$ and $^{13}C$ assimilation rates as well as the quantity of $^{13}C$ per gram dry weight of plant labeled, which is influenced by the quantity of $^{13}C$ added to the labeling chamber.

Sun et al. (2019), *PeerJ*, DOI 10.7717/peerj.7738

**Table 5 Intra-pot and inter-pot variation in $\delta^{13}$C value and $^{15}$N excess (in atom %) of different winter wheat components.**

| | | | CV (%) of intra-pot | | | | | | | CV (%) of inter-pot | | | | | | |
| | | | Ear | Leave | | | Stem | | | Ear | Leave | | | Stem | | |
| | | | | Upper | Middle | Lower | Upper | Middle | Lower | | Upper | Middle | Lower | Upper | Middle | Lower |
|---|---|---|---|---|---|---|---|---|---|---|---|---|---|---|---|---|
| $\delta^{13}$C (‰) | Elongation | −N[†] | 25.6 | 28.4 | 31.0 | 76.4 | 9.4 | 42.8 | 29.1 | 41.5 | 26.0 | 19.2 | 59.0 | 30.9 | 49.8 | 22.8 |
| | | +N[‡] | 21.2 | 31.6 | 42.6 | 46.5 | 36.1 | 43.7 | 35.8 | 47.1 | 34.3 | 26.5 | 24.0 | 54.1 | 32.2 | 16.5 |
| | Grain filling | −N | 11.9 | 25.7 | 13.0 | 7.3 | 11.6 | 16.2 | 6.3 | 9.7 | 10.8 | 15.1 | 6.1 | 9.7 | 5.5 | 8.0 |
| | | +N | 13.2 | 16.6 | 24.3 | 15.8 | 19.5 | 31.5 | 21.1 | 60.2 | 52.5 | 70.9 | 39.0 | 18.3 | 50.0 | 3.8 |
| $^{15}$N excess (in atom %) | Elongation | | 15.2 | 14.5 | 11.1 | 14.3 | 13.8 | 8.6 | 16.7 | 1.7 | 17.1 | 23.0 | 18.9 | 14.9 | 20.3 | 23.4 |
| | Grain filling | | 18.1 | 7.7 | 13.6 | 9.4 | 10.2 | 12.3 | 11.0 | 1.3 | 1.5 | 2.8 | 0.1 | 6.2 | 0.8 | 2.3 |

Notes.

[†] −N = no N addition.

[‡] +N = N addition.

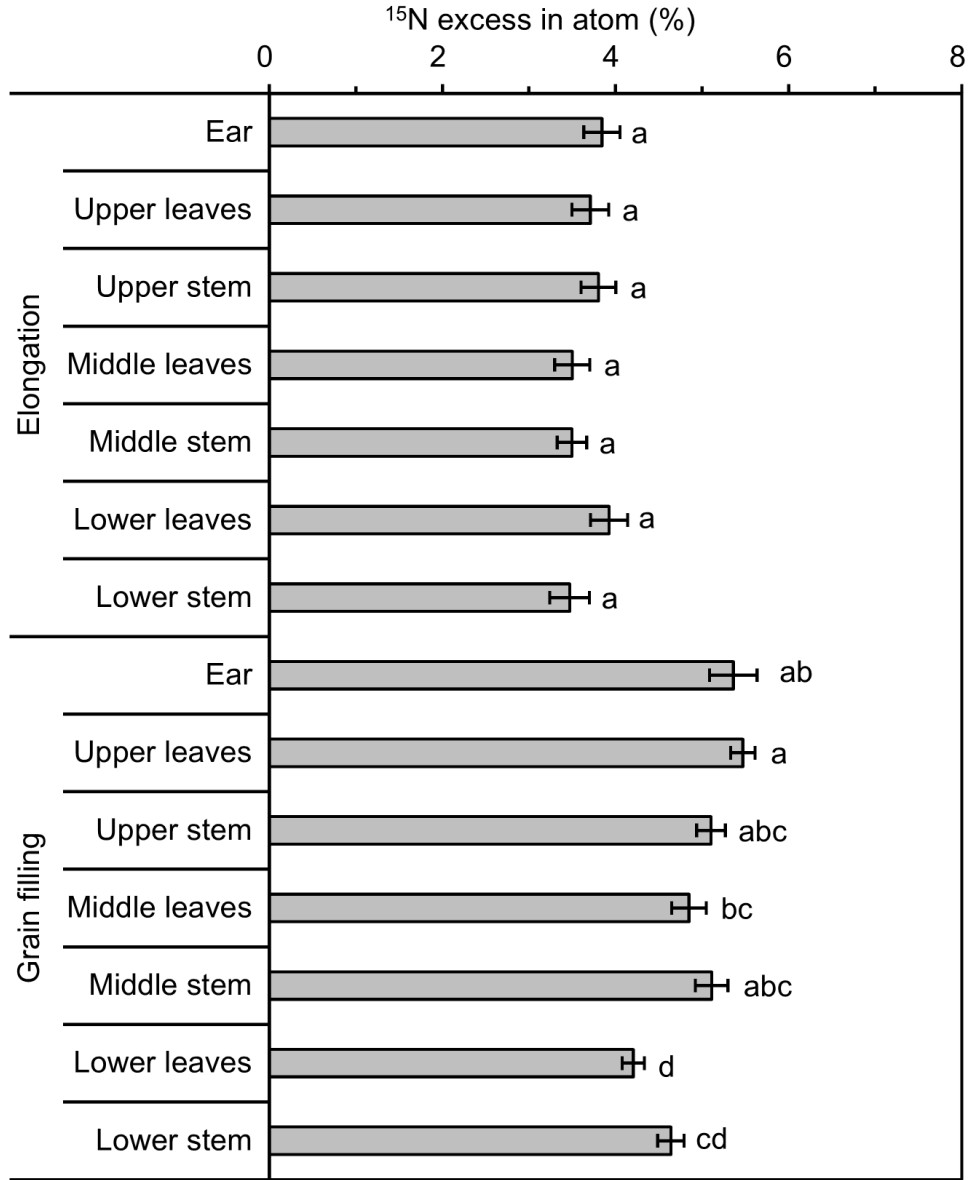

**Figure 4** **$^{15}$N excess (in atom %) of different winter wheat components with $^{15}$N labeling.** Different lowercase letters denote a significant difference among components at the same developmental stage (LSD, $p < 0.05$). Error bars indicate the standard error ($n = 9$).

The development of the winter wheat ear during the entire growth season was reflected in the increasing allocation of $^{13}$C to that reproductive organ. As growth advanced, $\delta^{13}$C values of ears significantly increased, whereas those of leaves and stems significantly decreased (Tables 1 and 2). This pattern, i.e., an increase in the carbohydrate sink strength during plant development, is typical of cereal crops. In a $^{14}$C pulse-labeling experiment, *Swinnen, Van Veen & Merckx (1994)* observed that as much as 85% of the $^{14}$C allocated to aboveground parts of spring wheat plants was recovered in the ears at the ripening stage.

In the present study, the proportion of $^{14}$C in ears relative to all aboveground portions increased from 30% at the elongation stage to 50% at the grain filling stage (Fig. 2A). Among aboveground components, $\delta^{13}$C values were also highest in ears at the grain filling stage (Table 2). At the grain filling stage, the ears were conditioned by the size and availability of the C pools. At the grain filling stage, the predominant C sources for wheat ears are primarily from remobilization of labeled photosynthesized C from leaves and stems, and from photosynthesis by the ear itself (*Palta et al., 1994*; *Aranjuelo et al., 2013*; *Zhou et al., 2016*). Previous studies that used the natural abundance of $^{13}$C and $^{13}$C-labeling techniques revealed that photosynthesis in the ear may account for 40%–80% of the C accumulated in the ears (*Palta et al., 1994*; *Aranjuelo et al., 2011*; *Sanchez-Bragado et al., 2014*). Given that the leaves and stems contained the lowest quantities of labeled C in wheat, these organs were indicated to act as a C source for grain filling, with ears as the sink containing higher amounts of labeled C. This result is in agreement with a previous C pulse-labeling study of wheat during grain filling (*Aranjuelo et al., 2013*). However, at the elongation stage, higher $^{13}$C-enriched values in the shoots (leaves and stems) were observed in the present study (Table 2). These results suggested that shoots acted as major labeled-C sinks owing to the poor sink strength of ears at the elongation stage, whereas ears represented an important labeled-C sink that competed with the shoots as the C sink at the grain filling stage (*Palta et al., 1994*; *Aranjuelo et al., 2013*; *Zhou et al., 2016*).

At the elongation stage, N addition tended to reduce $\delta^{13}$C values of aboveground components through a dilution effect (Table 2). Compared with those of the −N treatment, net $^{13}$C assimilation in plant biomass increased by 7.8% under the +N treatment (76 mg $^{13}$C pot$^{-1}$ for −N vs 82 mg $^{13}$C pot$^{-1}$ for +N), while the +N treatment showed a 35% increase in plant biomass (Fig. 2A). An additional likely relevant factor was reduced carbohydrate accumulation under over-supplied N conditions, i.e., under the +N treatment in the current study (*Palta et al., 1991*; *Gooding & Davies, 1992*). At the grain filling stage, N addition tended to increase $\delta^{13}$C values of aboveground components (Table 2). This result probably reflects the presence of a higher number of green leaves in the +N treatment at an advanced stage of maturity (grain filling), which leads to stronger photosynthetic capacity and hence can affect the $^{13}$C-isotopic signature of aboveground components.

At both labeled stages, $^{13}$C enrichment was higher in upper (newly developed) leaves than in lower (aged) leaves (Fig. 3), which may be explained by the more rapid uptake of $^{13}$CO$_2$ by younger leaves. Leaf age influences photosynthetic capacity and the $^{13}$C-isotopic signature (*Girardin et al., 2009*). In addition, young expanded leaves provide the majority of labeled C assimilates for stem growth (*Ryle & Powell, 1972*; *Subrahmanyam & Rathore, 2004*). Additional factors, such as light intensity (*Ryle & Powell, 1972*; *Subrahmanyam & Rathore, 2004*), air temperature (*Meharg & Killham, 1989*), and relative humidity (*Farquhar, O'leary & Berry, 1982*), also influence the photosynthetic rate and alter the C-isotopic signature. Wheat leaves showed higher $\delta^{13}$C values than those of stems (Table 2), which is consistent with previous $^{13}$CO$_2$ pulse-labeling studies, such as a study of grain sorghum (*Berg et al., 1991*). Leaf and stem positions (upper, middle, and lower) also influence the $\delta^{13}$C values of aboveground components (*Thompson, 1996*). In the present study, the upper leaves tended to be more enriched in $^{13}$C compared with that of the upper stems, whereas the lower leaves

were generally less $^{13}$C-enriched than stems at the same position (Table 2) (*Thompson, 1996*). This result was due to the remobilization of photosynthate and translocation of higher amounts of recently fixed C from leaves to stems during reproductive (grain filling) and vegetative (elongation) developmental stages. The strong variation in $\delta^{13}$C values in leaves and stems indicated that a single $^{13}$CO$_2$-labeling pulse did not produce uniformly $^{13}$C-labeled plant material. Repeat-pulse labeling has been suggested to overcome the technical constraints of continuous labeling and the high variation of the single-pulse labeling approach, and may generate adequately homogeneous $^{13}$C-labeled plant material (*Roper et al., 2013*; *Tahir et al., 2018*).

With respect to isotope signals, labeled plants also exhibited different patterns of variation among individual plants within a pot and among the three replicate pots (Table 5). Approximately 75% of the leaf and stem components had a CV for intra-pot $\delta^{13}$C higher than 30% at the elongation stage, whereas 67% had a CV less than 20% at the grain filling stage. This result demonstrates that the variability between individual wheat plants at the grain filling stage was smaller than that at the elongation stage, especially for the −N treatment. The variation in $^{13}$C-enrichment in different portions of the shoot at the elongation stage were markedly higher than those at the grain filling stage, which may reflect that young leaves are the most frequently labeled organs in wheat (*Girardin et al., 2009*; *Nguyen Tu et al., 2013*). The stronger labeling of starch is associated with its rapid synthesis during photosynthesis in leaves compared with that of stems during vegetative growth stages. The redistribution of labeled C from source organs (leaves and stems) to sink organs (ears), consisting of C assimilated during the grain filling period (*Aranjuelo et al., 2013*; *Zhou et al., 2016*), may lead to a decreased extent of variation in $^{13}$C enrichment among plant organs. The difference in variation patterns between the +N and −N treatments is likely because C and N remobilization starts earlier and is stronger when plants are grown under low N fertilization compared with the high N fertilization condition (*Aranjuelo et al., 2013*). Approximately 83% of leaf and stem components had an inter-pot $\delta^{13}$C CV higher than 20% at the elongation stage, whereas 67% had an intra-pot $\delta^{13}$C CV lower than 20% at the grain filling stage (Table 5). This result might indicate that the photosynthesized C in the leaves and stem is translocated to many destinations at the elongation stage, whereas the reallocation of labeled C from the leaves and stem is predominantly directly to the ears at the grain filling stage (*Palta et al., 1994*; *Aranjuelo et al., 2013*; *Zhou et al., 2016*). We therefore suggest that labeling of winter wheat at the grain filling stage without N fertilization may produce more uniformly $^{13}$C-labeled materials, whereas labeling of winter wheat at the elongation stage without N fertilization may produce more highly $^{13}$C-enriched plant materials, albeit with greater variability.

### Variation in $^{15}$N enrichment

Compared with that under $^{13}$C labeling, differences in $^{15}$N excess between aboveground components under $^{15}$N labeling conditions were much smaller (Table 2). Variation among individual plants (intra-pot) and among different pots (inter-pot) was also much lower (Table 5). No significant differences in $^{15}$N excess were observed among wheat aboveground components; the greatest difference, between ears and leaves, was only 11% at the grain

filling stage (Table 2). This result suggests that the $^{15}$N-enriched urea application used in this study can produce relatively uniformly labeled straw tissues at an advanced (grain filling) developmental stage in wheat. Nitrogen is a highly mobile nutrient in the plant, whereas certain other nutrients (i.e., C) are not translocated so readily within the plant. This outcome reflects that N is utilized as a nutrient by plants, whereas C is a basic element of plant photosynthesis. In contrast to plant $^{13}$C labeling, which requires complex environmental conditions, such as temperature, humidity, and $^{13}$CO$_2$ concentration, to be maintained throughout the labeling period (*Soong et al., 2014*), $^{15}$N labeling is relatively easy to achieve by means of $^{15}$N fertilization. Ideally, studies of the decomposition rate of straw tissue should be conducted, both to select uniformly enriched materials (*Crozier, King & Volk, 1993*; *Soong et al., 2014*) and to choose straw tissue similar to that of field-grown plant tissues (*Crozier, King & Volk, 1993*). The chemical composition of straw tissue at late maturity differs substantially from that at an early stage of maturity (*Niu et al., 2014*). For instance, the majority of plant N at advanced stages of maturity is associated with proteins, nucleic acids, and amino sugars (*Wagger, Kissel & Smith, 1985*; *Girardin et al., 2009*). Some nitrogenous components may exist in a form more resistant to microbial decomposition, such as N associated with cell wall material (*Wagger, Kissel & Smith, 1985*; *Girardin et al., 2009*).

As wheat growth progressed, i.e., with increase in biomass, $^{15}$N excess increased by 30.4%–41.1% (Table 2), which indicated that plant N demand was satisfied by available soil N derived from $^{15}$N fertilizer between the elongation and grain filling stages. Nitrogen demand per plant is predominantly satisfied through N uptake by roots prior to anthesis, and ear N content is augmented by the amount of N absorbed by the root system between anthesis and ear maturity (*Kichey et al., 2007*).

The $^{15}$N enrichment of shoots and ears differed among the developmental stages in winter wheat. At the elongation stage, no significant differences in $^{15}$N enrichment among the leaves, stems, and ears were observed. This result reflects that the redistribution of labeled N from source organs to ears was low and N in the heading ears was mainly absorbed from the soil during the vegetative growth stages (*Aranjuelo et al., 2013*). However, at the grain filling stage, wheat ears labeled with $^{15}$N tended to accumulate significantly higher quantities of $^{15}$N compared with other aboveground components (Table 2). Previous researchers have described similar phenomena following $^{15}$N labeling, such as in wheat (*Ladd, Oades & Amato, 1981*; *Palta et al., 1994*) and in winter wheat and grain sorghum (*Wagger, Kissel & Smith, 1985*). The ears are a strong N sink that affect N accumulation in the ears during grain filling, whereas the shoots and roots serve to supply nutrients to the ears (*Palta et al., 1994*; *Aranjuelo et al., 2013*; *Zhou et al., 2016*; *Sun et al., 2018*; *Zhou et al., 2018*). In the present experiment, 65% of the total N in grains was remobilized from the N accumulated in the shoots and roots during vegetative growth stages, whereas the remaining 35% was derived from N absorbed from the soil during the grain filling stage (*Sun et al., 2018*). Although ears exhibited significantly higher $^{15}$N enrichment compared with that of leaves and stems in winter wheat at the grain filling stage, the differences in $^{15}$N enrichment between leaves and stems were only slight (Table 2), which is suggestive of relatively uniform labeling in the plants because N accumulation and redistribution to

the grain occurs many days after N accumulation in the leaves and stems (*Butler & Ladd, 1985*; *Wagger, Kissel & Smith, 1985*).

## CONCLUDING REMARKS

Weight-average $^{13}C$ values for aboveground wheat components, resulting from $^{13}CO_2$ release from $Na_2{}^{13}CO_3$ (98 atom % $^{13}C$) in a field labeling chamber, ranged from 81‰ to 295‰. $^{13}C$ labeling of winter wheat at the grain filling stage without N fertilization may produce more uniformly $^{13}C$-labeled materials, whereas the materials were more highly $^{13}C$-enriched at the elongation stage, although the $^{13}C$ values of the wheat materials showed greater variability. The photosynthesized C in leaves and stems is transported to many destinations at the elongation stage, whereas at the grain filling stage reallocated C is predominantly translocated to the ears. The presence of a higher number of green leaves under the +N treatment than for the −N treatment at the grain filling stage reflected a stronger photosynthesis capacity and led to greater variation in the $^{13}C$-isotopic signature of aboveground components. Pulse labeling did not generate uniformly $^{13}C$-labeled plant materials. In contrast to $^{13}C$ labeling, which requires the control of a variety of environmental conditions throughout the labeling period, such as temperature, humidity, and $^{13}CO_2$ concentration, uniform $^{15}N$ labeling is relatively easier to achieve through $^{15}N$ fertilization; differences in isotope excess among aboveground wheat components were much smaller under $^{15}N$ labeling than under the $^{13}C$ labeling condition. Application of $^{15}N$-enriched urea in this study may produce more uniformly labeled wheat straw at the grain filling stage than at the elongation stage.

## ACKNOWLEDGEMENTS

We thank Zhao Zichao for his assistance with isotope analysis. We also thank the anonymous reviewers for their valuable comments.

### Funding

This study was financially supported by the National Key Research, the Development Program of China (grant no. 2016YFD0800100) and NSFC program (grant no. 31370527). The funders had no role in study design, data collection and analysis, decision to publish, or preparation of the manuscript.

### Grant Disclosures

The following grant information was disclosed by the authors:
National Key Research, the Development Program of China: 2016YFD0800100.
NSFC program: 31370527.

## Competing Interests

Biao Zhu is an Academic Editor for PeerJ. Roland Bol is employed by the Institute of Bio- and Geosciences, Agrosphere Institute (IBG-3), Forschungszentrum Jülich GmbH. The authors declare there are no competing interests.

## Author Contributions

- Zhaoan Sun performed the experiments, analyzed the data, prepared figures and/or tables, authored or reviewed drafts of the paper, approved the final draft.
- Shuxia Wu and Biao Zhu analyzed the data.
- Yiwen Zhang performed the experiments.
- Roland Bol authored or reviewed drafts of the paper.
- Qing Chen conceived and designed the experiments.
- Fanqiao Meng conceived and designed the experiments, analyzed the data, prepared figures and/or tables, authored or reviewed drafts of the paper.

## Data Availability

Raw data are available as Supplemental Files.

## Supplemental Information

Supplemental information for this article can be found online at http://dx.doi.org/10.7717/peerj.7738#supplemental-information.

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
