# Peer review of "Variation of 13C and 15N enrichments in different plant components of labeled winter wheat (Triticum aestivum L.)"

_PeerJ, doi:10.7717/peerj.7738_

## Round 0.1 · original submission · Major Revisions

The manuscript has been evaluated by three reviewers. Please carefully revise the manuscript incoporating the reviews from these reviewers.

·

Basic reporting

The article is well written in an understandable English

In my opinion there is more literature that should be commented in regard to d13C and d15N labeling and natural abundance. There is a lot of publications in wheat and other crops that studied C and N remobilization and measure natural abundance d13C and d15N, and use labeling with these two isotopes. See Zhou et al., 2014,2015, 2016 and papers published by Jose Luis Araus group, and Iker Aranjuelo (Aranjuelo et al., 2011; Plant Biology ISSN 1435-8603).

The article has a professional structure in figs, tables etc.. My only concern is that in the discussion there is a part in regarding to CV of d13C that seemed like a extension of the results. There is not real discussion with other publications. Please compare your results with other publications and try to find a sense to the different variation between high and low N treatments, and between the different organs. For example why the authors think that there is higher CV in the lower leaves than in the upper leaves? supported with other references.

In my opinion, the hypothesis of the manuscript is not clear enough or too descriptive, I will explain myself. Isotope labeling is a very powerful tool, to study C and N cycles, and has been used to study N2 fixation (Sanz-Saez et al., 2015), measure sink strength between organs and different cultivars (Zhou et al., 2014, 2015, 2016), N use efficiency (Felini et al., 2007), etc. However, in this study the author´s only objective is to describe the variation of 13C and 15N, not linking the labeling to a physiological process. There is no hypothesis or the intention understand a physiological process. In fact, there is no a reason why th authors are interested on describing the isotope labeling at different stages or organs. I think this part needs to be improved so the manuscript has a focus or a clear objective with physiological meaning.

Experimental design

The manuscript fulfill the scope of the journal

The research question is not well defined. See coments in point 1.

Comments to the Methods:
Please describe more in depth the 15N labeling procedure. Which was the N source for the labeling. How much was the richness of the 15N portion. How much 15N enrich fertilizer was added? please clarify.

The 15N labeling experiments need to be clarified. For what I understand (and I might have understood it wrong) the 15N treatments is only applied in the +N experiment, there is not +N that is not labeled control. I think that by only adding the 15N in the +N treatment, the authors may be mixing effects. To avoid this problem, the authors should have added an extra treatment in which they should have added a +N treatment but without labelling. I think that is the way to calculate the excess 15N value. In my opinion the N excess value as it is calculated now mix 15N and +N effects, so it is not completely correct. The authors should show the total N content (usually it is analyzed in the same 15N analysis) and in that way they will see how the N in the +N is much more concentrated. In that way authors would be able to calculate the difference in the 15N labeling between -N and +N to calculate the excess 15N in a more accurate way.

Did the authors take soil samples to analyze the 15N in the soil at the beginning of the experiment and at the end? That would help to see how much different the two N treatments were in the d15 signature.

The sample after the d13C labeling is at 28 days after labeling (DAL), can the authors explain why they choose those days. It seems to me that 28 DAL is too long to wait to analyze the labeling, the C13 can be respired in the majority and only a small proportion is still stored in the plant as starch or as structural sugars (see Aranjuelo et al., 2011 and Nogues et al., 2006).
In addition, a labeling of 7 hours for only one day seems a little insufficient to sample after 28 days, please support information that justify that methodology and maybe a reason, like the researcher wanted to study to which storage or structural molecules the C goes after respiration etc.

Can the author provide data about the 13C content in the chamber during the measurements? This is important because if there is not controlled and stable d13C signal that could explain the very high CV.

Justify why the authors separated the plant (stems) in three lengths. Is there any previous publication or justification for this? Use that in the introduction

Validity of the findings

In the discussion the authors speak of increase in biomass of 35% at high N but they don´t provide any data besides the d13C excess. Please provide the data. It would be useful to provide the data of leaves, stems, spike etc.to have more of a broad view of the data.

pp233-238. The authors suggest that the higher allocation of C to ears can be caused due to allocation from roots too ears. However the authors don´t continue the hypothesis or give any additional and original data to support the hypothesis, so I suggest to eliminate this part or say that is a speculation only based in data from other publications.

The part of the conclusion regarding to d13C does not clarify the source of variation in d13C. Please state all possible sources of variation like differences in the labeling, different leaf areas, photosynthesis.

Reviewer 2 ·

Basic reporting

The research titled as “Variation of 13C and 15N enrichments in different plant components of labeled winter wheat (Triticum aestivum L.)” tried to figure out the impact of labelling with 13C and 15N on two differenced developmental stages. The main innovation in this work is evaluate the labeling times to achieve a uniform labelled material. However, the results are not innovative. In addition, the text should be improved in English language as in some points that I describe below.
I suggest that do not use “trend” in a research work, and less when there are significant differences among your treatments.

Experimental design

The section Material and methods requires a higher explanation about the experimental set up.
I suggest explain the experimental design at the beginning of the “Treatments” section (i.e. total number of plots, number of plots per treatment and number of plants per plot).
L81. Replace CK by –N.
L82. You indicate that each treatment included four replications but until the line 97 do not indicate the number of plants per plot. I suggest group that.
L83. Did you apply 15N urea since the beginning of the experiment? Did you replace the 100% of N-fertilizer by 15N? Did you have wheat plants fertilized with N but non-labelled?
L102. You splitted the N fertilizer for the grain filling harvest. Plants that only received 1 doses of fertilizer also received 90 mg N hg-1 air-dried soil?
L117-122. I suggest extend this part for a clearly understanding. You indicate that for 13C labelling you chose 4 –N and 4 +N pots and 8 C-unlabeled controls. Were these controls 4 –N and 4 +N?
L145. Here you indicated that 3 plants per pot were assessed for the 13C and 15N enrichment and in line 82 you indicate that each treatment had 4 replications. So, you had 4 biological replicates with 3 technical replicates in each pot. Therefore, you had n=4 or 3 and you made 3 technical replicates for each treatment. Tables 1 and 3 and figures 1 and 2 indicate that the values are the means of 9 replicates. What do you consider as biological replicate and what as technical replicate?
L156. Did you used non-labelled material to calculate the 15N excess?
L178. I suggest indicate with words better than with symbols (next appearances as well)
L182. N fertilization decreased 13C values….
L184. N fertilization did not change the 13C values at grain filing stage (no significant differences).
L190-192. Upper stem did not present higher 13C values than middle stem. Here, the relevant result is that ear showed the highest 13C contents followed by upper leaves, which indicate the strong remobilization process.
L192-194. Are you meaning at grain filling stage? At elongation stage, there were not significant differences between upper and middle portions.

Validity of the findings

Your discussion needs more detail. I suggest that you discuss more the importance of ears at grain filling stage due to its importance as sink. Your results showed a high labelling of this component that could affect the other components by a strong remobilization process.
Results must be revised carefully and write that statistic indicates.
Conclusions are well stated and limited to supporting results.
The question of what do you consider biological replicate must be answered.

Reviewer 3 ·

Basic reporting

This manuscript describes the methods of artificially labeling outdoor-grown plants with 13C and 15N, and presents data on the resulting 13C and 15N enrichment in various plant parts.
Stable isotopic techniques may provide powerful insight to coupled soil-plant processes, and future advances in our understanding of the fate of plant C and N entering the soil will depend on efficient techniques which help to elucidate rather than obfuscate C and N cycling through the soil system. The authors seem to be fully aware of the importance of a uniform isotope distribution among the plant parts and chemical fractions being labeled to prevent misinterpretations.

The analysis of isotope homogeneity is a worthwhile contribution, but the results indicate that there was considerable inhomogeneity.

More serious is the failure to discuss several critical variables, including labeling frequency and chamber environmental conditions (temperature, humidity).

Experimental design

The experimental design is fine and clear.

Report the source and cost (in US dollars or Euros) of the NaHCO3 at 98 atom%

Validity of the findings

The results are clear and comprehensive however the proposed labeling method was marginally adequate, and produced residues that were marginally acceptable for subsequent decay studies, because 13C was not uniformly distributed among the plant parts. There is much room for improvement, and ideas to improve the labeling procedure should be shared openly.

The labeling intensity (duration of each pulse and frequency of pulses) may have a more important influence on above-ground residue quality and on isotopic homogeneity.

A more transparent discussion of strengths and weaknesses would add credibility and utility. Perhaps the authors might identify some mistakes, and advise readers on how they might be avoided. Actually this study provides very little support for conducting labeling in the field, the main benefits being ample PAR and a natural root environment and distribution.

Please see Tahir et al (2018) Field 13C Pulse Labeling of Pea, Wheat, and Vetch Plants for Subsequent Root and Shoot Decomposition Studies. Rev. Bras. Ciênc. Solo vol.42

Additional comments

The work has clear objectives and was carried out with rigor. However, the originality is limited, being a reapplication of the technique, as quoted in the manuscript.

---

## Round 0.2 · Minor Revisions

The revised manuscript has been checked by one of previous reviewers. This reviewer asked for a check on the English language. So I consider it to be acceptable only after a polish of the manuscript.

·

Basic reporting

The English should be checked before publication to polish some phrases, but otherwise is a well written manuscript.

Experimental design

All the suggested changes have been included so the experimental design is now acceptable.

Validity of the findings

THe novelty of the findings is not very high, but the results can be valuable for readers.

Additional comments

The manuscript has been considerably improved, but English grammar should be re-checked prior publication.

---

## Round 0.3 · accepted · Accept

I recommend an acceptance of the revised manuscript.